# The VertiGO! Trial protocol: A prospective, single-center, patient-blinded study to evaluate efficacy and safety of prolonged daily stimulation with a multichannel vestibulocochlear implant prototype in bilateral vestibulopathy patients

**Bernd L. Vermorken**[ORCID][1]*, **Benjamin Volpe**[1], **Stan C. J. van Boxel**[1], **Joost J. A. Stultiens**[1], **Marc van Hoof**[1], **Rik Marcellis**[ORCID][2], **Elke Loos**[1,3,4], **Alexander van Soest**[1], **Chris McCrum**[ORCID][2], **Kenneth Meijer**[ORCID][2], **Nils Guinand**[1,5], **Angélica Pérez Fornos**[ORCID][5], **Vincent van Rompaey**[ORCID][1,6,7], **Elke Devocht**[ORCID][1], **Raymond van de Berg**[1]

1 Department of Otorhinolaryngology and Head and Neck Surgery, Division of Balance Disorders, School for Mental Health and Neuroscience (MHENS), Maastricht University Medical Centre, Maastricht, The Netherlands, 2 Department of Nutrition and Movement Sciences, NUTRIM School of Nutrition and Translational Research in Metabolism, Maastricht University, Maastricht, Netherlands, 3 Department of Neurosciences, Research Group Experimental Oto-Rhino-Laryngology (ExpORL), KU Leuven, University of Leuven, Leuven, Belgium, 4 Department of Otorhinolaryngology-Head and Neck Surgery, University Hospitals Leuven, Leuven, Belgium, 5 Division of Otorhinolaryngology Head and Neck Surgery, Department of Clinical Neurosciences, Geneva University Hospitals, Geneva, Switzerland, 6 Faculty of Medicine and Health Sciences, University of Antwerp, Antwerp, Belgium, 7 Department of Otorhinolaryngology and Head and Neck Surgery, Antwerp University Hospital, Edegem, Belgium

* bernd.vermorken@mumc.nl

# Abstract

## Background

A combined vestibular (VI) and cochlear implant (CI) device, also known as the vestibulocochlear implant (VCI), was previously developed to restore both vestibular and auditory function. A new refined prototype is currently being investigated. This prototype allows for concurrent multichannel vestibular and cochlear stimulation. Although recent studies showed that VCI stimulation enables compensatory eye, body and neck movements, the constraints in these acute study designs prevent them from creating more general statements over time. Moreover, the clinical relevance of potential VI and CI interactions is not yet studied. The VertiGO! Trial aims to investigate the safety and efficacy of prolonged daily motion modulated stimulation with a multichannel VCI prototype.

## Methods

A single-center clinical trial will be carried out to evaluate prolonged VCI stimulation, assess general safety and explore interactions between the CI and VI. A single-blind randomized controlled crossover design will be implemented to evaluate the efficacy of three types of

**Data Availability Statement:** No datasets were generated or analysed during the current study. All relevant data from this study will be made available upon study completion.

**Funding:** B.L.V., E.D. and A.v.S. were and will be supported through funding of MED-EL and ZonMw. B.V. and S.C.J.v.B. were and will be supported through funding of MED-EL, Heinsius Houbolt fonds and Stichting De Weijerhorst. E.L. is funded by the Klinische onderzoeks- en opleidingsraad (KOOR) of the University Hospitals Leuven. N.G., A.P.F and R.B. received several research and travel grants from MED-EL. The funders had no role in study design and will not have a role in data collection, data analysis, interpretation of data, decision to publish, or preparation of the manuscript. The remaining authors declare that research will be conducted in the absence of any commercial or financial relationship that could be construed as a potential conflict of interest.

**Competing interests:** I have read the journal's policy and the authors of this manuscript have the following competing interests: the current study willl be financially supported by the Dutch Government (ZonMw, Health~Holland grant number 40-44600-98-330), Foundation "Stichting Het Heinsius-Houbolt Fonds", MED-EL (Innsbruck, Austria) and "Stichting De Weijerhorst".

**Abbreviations:** ABR, Auditory brainstem response; AE, Adverse event; aECAP, Auditory electrically evoked compound action potential; AMP, Audio-motion processor; ART, Audio response telemetry; BV, Bilateral vestibulopathy; BVQ, Bilateral vestibulopathy questionnaire; CAREN, Computer assisted rehabilitation environment system; CBCT, Cone beam computed tomography; CI, Cochlear implant; CNC, Consonant-nucleus-consonant; cVEMP, Cervical vestibular evoked myogenic potentials; DHI, Dizziness handicap inventory; DiN, Digit in noise; DQ, Daily questionnaire; DR, Dynamic range; DVA, Dynamic visual acuity; ECAP, Electrically evoked compound action potential; EQ5D-5L, Euroqol five-dimensional questionnaire; FES-I, Falls efficacy scale international; fHIT, Functional head impulse test; HADS, Hospital anxiety and depression scale; HUI-3, Health utility index mark 3; ICECAP-A, Icepop capability measure for adults questionnaire; Mini-BESTest, Mini-balance evaluation systems test; MUMC+, Maastricht University Medical Centre+; OSQ, Oscillopsia severity questionnaire; oVEMP, Ocular vestibular evoked myogenic potentials; PAM, Pulse amplitude modulation; PRAM, Pulse rate + amplitude modulation; PROM, Patient-reported outcome measure; PSFS, Patient-specific

stimulation. Furthermore, this study will provide a proof-of-concept for a VI rehabilitation program. A total of minimum eight, with a maximum of 13, participants suffering from bilateral vestibulopathy and severe sensorineural hearing loss in the ear to implant will be included and followed over a five-year period. Efficacy will be evaluated by collecting functional (i.e. image stabilization) and more fundamental (i.e. vestibulo-ocular reflexes, self-motion perception) outcomes. Hearing performance with a VCI and patient-reported outcomes will be included as well.

## Discussion

The proposed schedule of fitting, stimulation and outcome testing allows for a comprehensive evaluation of the feasibility and long-term safety of a multichannel VCI prototype. This design will give insights into vestibular and hearing performance during VCI stimulation. Results will also provide insights into the expected daily benefit of prolonged VCI stimulation, paving the way for cost-effectiveness analyses and a more comprehensive clinical implementation of vestibulocochlear stimulation in the future.

## Trial registration

ClinicalTrials.gov: NCT04918745. Registered 28 April 2021.

## 1 Background

### 1.1 Bilateral vestibulopathy

Bilateral vestibulopathy (BV) is defined as a chronic severe loss of function of both balance organs (quantified by a significantly impaired or absent function of the vestibulo-ocular reflex), which represents a major handicap involving strong balance disturbances, higher risk of falling, oscillopsia (i.e. a symptom of blurred vision during head movements), and associated loss of autonomy and quality of life [1–4]. About three million people globally are believed to be affected by bilateral vestibulopathy, but this is most likely an underestimation due to misdiagnosis [5]. At this point, only limited rehabilitation strategies are possible, focusing on compensation, adaptation and substitution [6], instead of treatment or restoration of vestibular function.

### 1.2 The vestibular implant

The concept of a vestibular implant (VI) was conceptually proven in animal studies [7–9]. The VI is a neuroprosthesis that aims to restore vestibular function. Similar to a cochlear implant (CI), the dysfunctional sensory input in the vestibular end organ can be bypassed by directly stimulating the vestibular nerves [10]. Head motion is sensed by gyroscopes and accelerometers. A processor converts this head motion information into electrical signals. The implanted stimulator transmits these signals to the vestibular (ampullary) nerves [10] via electrodes implanted in the vicinity of the neural targets.

Currently, different VI prototypes are being evaluated regarding feasibility and functionality by several research groups worldwide [11, 12, 13–21]. Since hearing can be (further) compromised when the labyrinth is opened during surgery [22], a combined VI and CI prototype was developed, referred to as a vestibulocochlear implant (VCI), to restore vestibular function

functional scale; QALY, Quality adjusted life years; SAE, Serious adverse event; SCC, Semicircular canal; SSQ-12, Speech spatial and qualities of hearing scale; T, Threshold; TQ, Tinnitus questionnaire; UCL, Upper comfortable level; UZA, Antwerp University Hospital; VA, Visual acuity; VAS, Visual analog scale; VBR, Vestibular brainstem response; VCI, Vestibulocochlear implant; vHIT, Video head impulse test; VI, Vestibular implant; VOG, Vestibulo-oculography; VOR, Vestibulo-ocular reflex; VRT, Vestibular rehabilitation therapy.

and hearing function concurrently. The first VCI prototypes containing one vestibular electrode contact were implanted in humans in 2007. These prototypes evolved into multichannel devices, implanted for the first time in humans in 2012. These multichannel devices contain multiple vestibular branches with on top of each branch one electrode contact. In the last decade, multiple promising results involving VI stimulation were reported, including a (partial) restoration of the vestibulo-ocular reflex (VOR) [15, 16]. This restoration resulted in an improved dynamic visual acuity, which has the potential of restoring gaze stabilization [19, 23, 24]. Additionally, natural frequency-dependence of VI stimulation was demonstrated [17] and eye movement alignment was investigated [25]. These studies show that subjects tolerate and adapt to VI stimulation and that electrically evoked VOR (partly) mimics responses also seen in the "normal" vestibular system. Moreover, postural responses were elicited and influenced by electrical stimulation [26, 27]. Improvements in postural responses and gait were described in patients receiving stimulation of the vestibular organ [28].

Until now most studies investigated the acute and intermittent use of a VI. Recently, a few research groups started to evaluate chronic effects of semicircular canal stimulation [25, 29] and otolith stimulation [30]. These groups showed stable restoration of the VOR over time and positive subjective outcomes. There were no adverse effects of prolonged continuous daily stimulation with a VI-only device [25]. However, based on these studies alone, with limited patient numbers and specific etiologies, it is not possible to generalize the results to a larger patient population or create more general statements of rehabilitation prospects in the long term. To the best of our knowledge, randomized controlled trials (RCTs) measuring VI performance in a number of strategies are currently nonexistent. Most publications focus on the efficacy of one fitting paradigm, in order to compare these settings with an OFF-set mode. To the best of our knowledge, there are no studies comparing a number of different stimulation modes based on different fitting parameters. Moreover, long periods of continuous stimulation are at this point only investigated with a VI-only device, and not yet with a VCI. Therefore, there is a need to investigate the safety and efficacy of prolonged stimulation with a VCI in a randomized controlled trial setting.

### 1.3 The VertiGO! Trial

To take vestibular implant research a step further, a new clinical trial will be conducted in a specific patient population suffering from both BV and severe sensorineural hearing loss. This trial primarily focuses on the overall safety and efficacy of prolonged VCI stimulation. A new refined multichannel VCI prototype will be evaluated, including concurrent multichannel VI and CI stimulation. Furthermore, this study aims to provide fundamental insights into the programming (fitting) of future VI devices by comparing to the condition without VI stimulation to three different modes of VI stimulation: (A) motion-modulated stimulation with baseline stimulation, (B) motion-modulated stimulation with reduced baseline stimulation, and (C) baseline stimulation (no modulation). A schematic representation of these three modes of stimulation is shown in Methods, paragraph 2.2.5. Baseline stimulation is defined as a continuous constant-amplitude electrical pulsatile signal, mimicking the "resting" activity of the vestibular nerve. Baseline stimulation is not simply considered as a control condition in this trial. Prolonged unmodulated stimulation might enhance the residual vestibular function (including the potential effects of stochastic resonance) [11]. Moreover, this unmodulated strategy is already applied in studies investigating the potential effect of otolith stimulation [30] and in studies investigating the effect of galvanic vestibular stimulation [31]. In these studies, the constant pulse train of baseline stimulation is believed by some researchers to mask the absence of physiological stimuli in BV patients [30].

By comparing two types of prolonged motion-modulated stimulation with different levels of baseline stimulation, it is expected to further refine the potential efficacy of vestibular stimulation. The potential efficacy of every stimulation paradigm will be evaluated and compared in terms of maximum effectivity, tolerance and compliance, flexibility and sustainability. A sham-like stimulation, in the form of 'reversed' stimulation, will not be applied due to unwanted added burden for the participants (e.g. significant prolongation of study interventions). Secondary goals involve exploring potential interactions between simultaneous CI and VI stimulation, overall auditory performance with the device (CI performance), and long-term follow-up of the VCI-implant over a period of 5 years. This study also aims to collect data on patient experiences and perspectives as input for early health technology assessments. Furthermore, this trial will serve as a proof of concept for a VI-fitting program and a VI-specific rehabilitation program. Finally, different stimulation parameters (i.e. modulation types and transfer functions) and more fundamentally central responses to VI stimulation will be evaluated.

Four hypotheses are addressed in the VertiGO! Trial:

1. Prolonged stimulation with a VCI is safe and effective; VI stimulation will result in improved dynamic visual acuity, increased postural control, higher VOR gain values and fewer BV-related symptoms compared to no VI stimulation.

2. Motion-modulated stimulation with 50% or reduced baseline stimulation will result in improved dynamic visual acuity, increased postural control, higher VOR gain values and fewer BV-related symptoms compared to baseline stimulation alone. Furthermore, it is expected that the level of baseline stimulation influences VOR gain resulting in higher but asymmetric VOR gain values and improved dynamic visual acuity in situations with reduced baseline stimulation.

3. Responses to VCI stimulation are expected to change during the first days of stimulation, since most of the adaptation to misalignment that might be present between the actual head rotation and the electrically evoked VOR is expected to be compensated for during the first week of stimulation [32]. Alongside the potential VOR directional plasticity, it is likely that other adaptation processes would occur during the first weeks of stimulation, potentially influencing the stimulation efficiency and induced responses. This is already a known phenomenon in CI-only implementation [33]. Therefore, it is expected that outcomes will improve throughout each prolonged stimulation period. The effect of acute stimulations are expected to stay stable during the whole follow-up period.

4. VI stimulation is expected to interact with CI stimulation and vice versa. It is expected that these bidirectional effects on vestibular or hearing performance are not clinically relevant.

## 2 Methods/ design

### 2.1 Design, setting and recruitment

A single-center clinical trial will be performed in the Maastricht University Medical Centre+ (MUMC+), a tertiary university medical center, in cooperation with Geneva University Hospitals. The proposed design will be a single-blind randomized crossover controlled trial. The protocol is approved by the local medical ethical committee (MUMC+, NL73492.068.20/ METC 20–087), is registered at clinicaltrials.gov (NCT04918745) and will be conducted in accordance with the Declaration of Helsinki. All subjects included in this study were informed by a medical doctor, which is part of the research team and have signed an informed consent

(S1 Appendix, Supporting information, model consent form) form before participation. The trial will be monitored by the local ethical committee, independent from investigators. The research protocol will be made available through the supplemental materials published with this article. Every single modification to the protocol will be communicated to the local medical ethical committee. In case a formal amendment to the protocol is required and approved, this amendment will be included in a new version of the informed consent prior to implementation according to the local regulations. The protocol with formal amendments will be made available together with the final trial reports after finalization of the trial. Modifications that do not require a formal amendment will be documented with version control. The SPIRIT reporting guidelines are used in reporting this trial (S2 Appendix, Supporting information, SPIRIT guidelines) [34]. See S3 Appendix, Supporting information, for the World Health Organization Trial Registration Data Set.

The trial aims to include a minimum of eight, with a maximum of 13, patients with diverse BV etiologies and different durations of disease, to represent a heterogeneous patient population. Recruitment started on 01 July 2021, will end in the first quarter of 2024, and takes place in the MUMC+. Every patient that is believed to suffer from a balance disorder or is candidate for cochlear implantation, according to their health records, undergoes a vestibular screening, as part of the standard clinical routine at the ENT department of the MUMC+. Patients who appear to be eligible for (and are interested in) participating in VI-related studies serve as the population base and will be personally informed by an MD about the VertiGO! Trial in the outpatient clinic during one of their regular follow-up visits at the ENT department. The time after diagnosis of vestibulopathy and/ or hearing loss at inclusion may vary. No advertisements will be utilized. Adequate enrolment will be promoted by actively monitoring outpatient clinics devoted to vestibular patients and potential CI candidates.

In order to be eligible to participate in this study, a subject must meet all of the following criteria, which are based on the recent opinion statement about vestibular implantation criteria [35]:

1. Be aged 18 years or older;

2. Suffer from disabling symptoms of postural imbalance and/ or oscillopsia;

3. Have reduced or absent bilateral VOR function based on at least one of the tests shown in Table 1 (meeting criteria A, with the other tests meeting criteria B of the vestibular implantation criteria statement [35]);

4. With an onset of bilateral vestibulopathy after the age of two;

5. With documented vestibular dysfunction from a peripheral origin;

**Table 1. Vestibular inclusion criteria.**

| VOR function test | Criteria A | Criteria B |
|---|---|---|
| Caloric test[1] | Each side $\leq 6$˚/sec | Each side $< 10$˚/sec |
| vHIT gain | Bilateral horizontal SCC $\leq 0.6$<br>1 Bilateral vertical SCC $<0.7$ | 2 Bilateral SCC $< 0.7$ |
| Rotatory chair[2] | Gain $\leq 0.1$ | Gain $\leq 0.2$ |

Each participant should meet at least one criteria A with the other tests meeting criteria B

[1] Sum of bithermal (30˚ and 44˚) maximum peak slow-phase velocity,

[2] Horizontal VOR function with sinusoidal stimulation on a rotatory chair at 0.1 Hz with Vmax = 50˚/sec

SCC (Semicircular canal)

6. Have severe to profound sensorineural hearing loss, meeting Dutch CI-candidacy criteria (in the ear to be implanted);

7. Documented post-lingual onset of profound hearing loss ($> 4$ years of age).

A subject is excluded when meeting one or more of the following criteria:

1. Radiographic evidence of absent/abnormal vestibular/cochlear end-organs impeding implantation;

2. Has any sign of central vestibular/cochlear dysfunction or structural vestibular/cochlear nerve pathology;

3. Has received a cochlear implant in the contralateral ear from another brand than the VCI brand (*MED-EL, Innsbrück, Austria*);

4. Is unwilling to stop the use of vestibular suppressant medications;

5. Shows signs of recovery of vestibular function, oscillopsia and imbalance within 6 months from the onset of symptoms;

6. Is not a proficient speaker of the Dutch language;

7. Presents any contra-indication for VCI surgery;

8. Shows non-vestibular pathologic conditions of sufficient severity to confound vestibular function tests;

9. Presents any contra-indication for MRI or CT imaging prior to surgery.

All patients who signed an informed consent document witnessed by a medical doctor and passed all screening tests, are identified by an anonymized 2-digit patient ID (CVI-xx).

## 2.2 Study interventions

A complete overview of study visits, including all scheduled procedures and outcomes per visit, is presented in Fig 1. A timeline of all study-related visits per subject is shown in Fig 2. Briefly, after inclusion, patients will be scheduled for VCI surgery. After implantation, subjects will follow a regular CI rehabilitation program with the addition of one postoperative control visit, a cone beam computed tomography (CBCT) scan and four hours of acute VI testing. Once the CI rehabilitation is finished, a four-day VI fitting period, one day of baseline testing and three periods of four days (three-period crossover design) of prolonged VI stimulation will be scheduled as the main trial experiments of this study outline. During each regular CI follow-up visit, some additional VI-related follow-up tests will be conducted. In total, a five-year follow-up period after the first implant fitting will be performed. Before every day of study interventions, the participant will be personally contacted by one of the researchers to explain all upcoming experiments of the next day, the importance of specific instructions and preparations will be underlined. One week after each prolonged stimulation week, the participant will be personally contacted to ask about any positive or negative effects they are experiencing after the intervention (part of the safety outcomes). Due to frequent contact moments between participants and researchers, the participant retention will be promoted.

A classical RCT with parallel groups multiplies the number of participants which is not feasible in this specific intervention. Moreover, it increases the risks of suboptimal treatment and outcomes in one of the groups. Therefore, we propose an alternative crossover study design that is feasible, takes a within-subject perspective, and in which the participants serve as their

| TIMEPOINT | Enrollment | VCI surgery | Postoperative visit | CI rehabilitation [†] | VI fitting | Reference testing | Prolonged VCI stimulation | VCI follow-up [†] |
|---|---|---|---|---|---|---|---|---|
| **Interventions** | | | | | | | | |
| VCI implantation | | X | | | | | | |
| CI fitting [†] | | | | X | | | | X |
| Speech therapy [†] | X | | | X | | | | X |
| Acute VI stimulation | | X | | X | X | | X (daily) | X |
| Prolonged VI stimulation | | | | | | | X (daily) | |
| Exercise program | | | | | | | X (2/week) | |
| **Assessments** | | | | | | | | |
| **Main primary outcomes** | | | | | | | | |
| DVA | X | | | | X | X | X (daily) | |
| Safety ((S)AE's) | | X | X | X | X | X | X | X |
| **Additional primary outcomes** | | | | | | | | |
| _Vestibular testing_ | | | | | | | | |
| fHIT | | | | | | X | X (2/week) | |
| 3D vHIT | X | | | | | X | X (2/week) | |
| Rotatory chair | X | | | | | X | X (2/week) | |
| Self-motion perception | | | | | | X | X (2/week) | |
| _Gait analysis_ | | | | | | | | |
| Mini-BESTest | | | | | | X | X (2/week) | |
| CAREN-testing | | | | | | X | X (1/week) | |
| **Secondary outcomes** | | | | | | | | |
| _Telemetry_ | | | | | | | | |
| Impedances (IFT) [†] | | X | | X | X | | X (daily) | X |
| ART (vECAP, aECAP [†]) | | X | | X | X | | | |
| _Central responses_ | | | | | | | | |
| ABR & VBR | | | | | X | | X | |
| _Auditory outcomes_ | | | | | | | | |
| Tone audiometry [†] | X | | | X | | | | |
| Speech recognition (CNC) [†] | X | | | X | X | X [#] | X (2/week) [#] | X |
| DiN | | | | X | X | X [#] | X (2/week) [#] | X |
| _Patient-reported outcomes_ | | | | | | | | |
| DQ | X | | | | | X | X (daily) | |
| BVQ, DHI, OSQ, HADS, FES-I | X | | | | | X | | |
| SSQ-12 [†] | X | | | X | | | | X (every visit) |
| TQ [†] | X | | X* | X* | | | | X* |
| EQ5D-5L, ICECAP-A, HUI-3 | X | | | | | X | X (1/week) | X (annually) |
| PSFS | | | | | | X | X (1/week) | |
| Semi-structured interview | | | | X | X | | X (1/week) | |
| _Imaging outcomes_ | | | | | | | | |
| CT [†] & MRI | X | | | | | | | |
| Fluoroscopy | | X | | | | | | |
| 3D imaging | | X | | | | | | |
| CBCT | | | X | | | | | X (12, 24, 60 months) |
| **Screening assessments** | | | | | | | | |
| oVEMP & cVEMP | X | | X | | | | | |
| Caloric test | X | | | | | | | |

**Fig 1. Schedule of enrollment, interventions and assessments per study visit per subject.** aECAP (auditory electrically evoked compound action potential), ART (audio response telemetry), BVQ (bilateral vestibulopathy questionnaire), CAREN (computer assisted rehabilitation environment system), CBCT (cone beam computer tomography), CI (cochlear implant), CNC (consonant-nucleus-consonant), CT (computer tomography), cVEMP (cervical vestibular evoked myogenic potentials), DHI (dizziness handicap inventory), DiN (Digits-in-Noise), DQ (daily questionnaire), DVA (dynamic visual acuity), FES-I (falls efficacy scale international), fHIT (functional head impulse test), HADS (Hospital anxiety and depression scale), HUI-3 (health utility index mark 3), ICECAP-A (ICEpop capability measure for adults questionnaire), mini-BESTest (mini-balance evaluation systems test), MRI (magnetic resonance imaging), OSQ (oscillopsia severity questionnaire), oVEMP (ocular vestibular evoked myogenic potentials), PSFS (patient-specific functional scale), SSQ-12 (speech spatial and qualities of hearing scale), TQ (tinnitus questionnaire), VBR (vestibular brainstem response). VCI (vestibulocochlear implant), vECAP (vestibular electrically evoked compound action potential), vHIT (video head impulse test), VI (vestibular implant). *If tinnitus complaints are higher at post-operative measurement, follow-up measurements will be performed. [†] Part of regular CI work-up. [#] Measured in two conditions: static and dynamic.

own control. For each subject, the three stimulation modes will be randomly assigned to one of three prolonged stimulation periods by using a block randomization-based design [36].

Since there are three stimulation modes which will be randomized over three time periods, there are six distinct algorithm sequences. To equally distribute all the patients over six

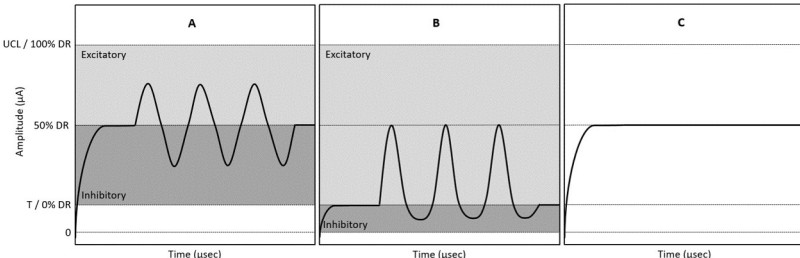

**Fig 2. Timeline per subject of the VertiGO! trial.** Timeline in months (m). VCI (vestibulocochlear implant), CI (cochlear implant), VI (vestibular implant).

algorithm sequences a block size of six is used. A custom "urn" will be created in the following way: Each of the six possible algorithm sequences is added to the urn once. Additional algorithm sequences are chosen at random without replacement from the total of six algorithm sequences and also added to the urn. This is done until the amount of algorithm sequences in the urn is equal to the amount of patients which will be included. The urn now contains as many allocation possibilities as there are patients, with each allocation possibility featuring at least once and no more than twice. Randomization will be performed in the period after VI fitting and before the prolonged stimulation periods. Previous experience with vestibular stimulation illustrated that turning on stimulation is clearly noticeable by both the patients and the examiners [12]. To decrease patient-induced bias during the prolonged stimulation periods, the patients will be blinded as to which stimulation mode is used in which week. Each stimulation mode is coded with a letter, code breaks to the patient should only occur in circumstances when knowledge of the actual stimulation mode is essential for further management of the patient. Although these circumstances seem highly unlikely, participants and/ or researchers who believe that unblinding is necessary, are encouraged to discuss this with the independent physician of this trial. In all other circumstances, researchers are encouraged to maintain blinding totally. All code breaks will be reported to the principal investigator. Unblinding will not be a reason for study discontinuation.

**2.2.1 VCI surgery.** CI implantation will be done using the round window approach. VI implantation will be performed using the intralabyrinthine approach, as previously described [13]. The electrodes will be inserted into the semicircular canals, with the aim to implant the electrode contacts close to the ampullary nerves. To optimize electrode placement, real-time fluoroscopy-guidance as described in [37] will be performed. After initial placement and after fixation, electrode placement will be verified by 3D images, fused with preoperative imaging samples (see 2.4.3.5). Once inserted, every VI electrode will be briefly stimulated in order to document electrically elicited eye movements during surgery.

**2.2.2 CI rehabilitation and acute VI stimulation.** Subjects will undergo a regular CI rehabilitation period, following the MUMC+ cochlear implant standard procedures. At seven, twelve and seventeen weeks (± one week) after implantation the vestibular electrodes will be acutely stimulated. By shortly stimulating each vestibular electrode with pulse trains, the dynamic range (DR) available for stimulation will be determined in a stepwise procedure from threshold (T) up to the Upper Comfortable Level (UCL). These levels will be based on the quality and quantity (VAS-based) of perception, eye movements and/ or the occurrence of facial nerve stimulation. Concurrently electrically evoked VOR will be investigated by recording two-dimensional eye movements.

**2.2.3 VI fitting.** To elaborate on the individual VI settings, a 4-day fitting period will be scheduled. The DR will be determined per electrode, similar to earlier acute VI stimulations.

After the DR is determined, baseline stimulation will be activated. An adaptation period, as described in [38], is included, and its duration is based on nystagmus and perception. From the 'adapted' state, block and sinusoid modulations will be applied [39]. For sinusoidal modulations, a modulating frequency of 2 Hz is used as default, since it is known to be within the frequency range of head movements during daily activities, a VOR could previously be evoked at this frequency and this frequency is not likely to be simulated purposefully by subjects [39–42]. In case of facial nerve stimulation and/ or uncomfortable perception, the UCL will be adapted. Each VI electrode will be fitted separately in blocks of two hours. The fitting will be repeated on the next day to test for reproducibility, taking up a total of three days. Furthermore, the potential influence of varying stimulation- and modulation settings (e.g. pulse phase duration and stimulation frequency) on electrically evoked eye responses and patients' perception will be investigated. Once the fitting of the individual vestibular electrodes is completed, the final fitting day will focus on all three combinations of two vestibular electrodes and the combination of all three electrodes.

**2.2.4 Reference testing day.** To investigate the patient's vestibular performance without vestibular stimulation, a day will be scheduled which includes multiple vestibular tests as shown in Fig 1. These tests will be carried out without the use of the vestibular implant. The maximum time period between this day of testing and prolonged stimulation days will be kept short, to ensure that the results from reference testing serve as a good control for the prolonged stimulation periods.

**2.2.5 The prolonged stimulation period.** One prolonged stimulation period consists of four days of stimulation, eight hours a day. This schedule will be performed three times (Fig 3). The only parameter that will randomly change between each period is the used stimulation paradigm. The three stimulation paradigms are schematically shown in Fig 4. The parameters (e.g. level of baseline stimulation and modulation) of each paradigm will be determined after analyzing the results of VI fitting. The DR will be determined on the first day of every prolonged stimulation period, in order to verify the stability of results found earlier during acute stimulation and VI fitting. Most outcome measures will be measured at least twice per week, as described in Fig 1.

Apart from the wide range of experimental procedures, two sessions of a 60-minutes physiotherapeutic supervised exercise program will be scheduled. A personalized mixture of gaze stabilization exercises, habituation exercises and balance and gait training will be carried out.

| Day | 1 | 2 | 3 | 4 |
|---|---|---|---|---|
| Time | | | | |
| 09:00 — A | A | A | A | A |
| | Habituation | Habituation | Habituation | Habituation |
| 10:00 — | | Perception platform | CAREN | Perception platform |
| 11:00 — | Break | Break | Break | Break |
| | Head impulse tests | Head impulse tests | Exercise program | Mini-BESTest & PSFS |
| 12:00 — | Break | Break | Break | Break |
| 13:00 — | DVA | DVA | DVA | DVA |
| 14:00 — | Break | Break | Break | Break |
| 15:00 — | Rotatory chair | Exercise program | Rotatory chair | Dynamic CNC and DiN |
| 16:00 — | Break | Break | Break | Break |
| | Mini-BESTest & PSFS | Static CNC and DiN | Head impulse tests | Semi-structured interview |
| 17:00 — B | B | B | B | B |

**Fig 3. Example schedule of a prolonged stimulation period.** [A] Steady state habituation and short fitting check. [B] Impedance, UCL and VAS-based list of questions. It should be noted that actual timeslots may differ between subjects. CAREN (computer assisted rehabilitation environment system), CNC (consonant-nucleus-consonant), DiN (Digits-in-Noise), DVA (dynamic visual acuity), mini-BESTest (mini-balance evaluation systems test), PSFS (patient-specific functional scale), UCL (upper comfortable level), VAS (visual analog scale).

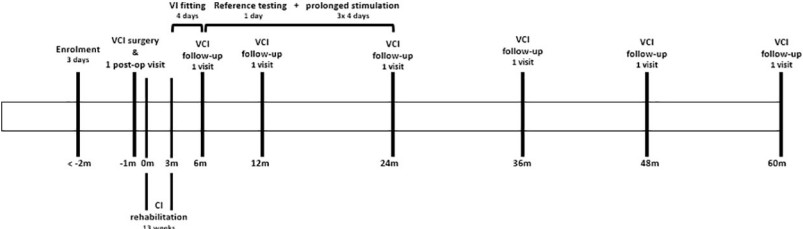

**Fig 4. Three modes of stimulation by the vestibular implant.** (A) Motion-modulated stimulation with baseline stimulation. (B) Motion-modulated stimulation with reduced baseline stimulation. (C) Baseline stimulation (no modulation). Baseline stimulation is given as a constant-amplitude electrical pustule signal in all three stimulation modes. Excitatory modulation is shown in light gray and inhibitory modulation is shown in dark gray.

## 2.3 Hardware and software

The VCI implant is a modification of a MED-EL Synchrony cochlear implant (PIN Mi1200 variant) with 12 electrode contacts (*MED-EL*, *Innsbruck*, *Austria*). In the VCI configuration, the three most basal electrodes are separated from the main cochlear electrode array into three individual branches (Fig 5A). These three individual branches are designed to be inserted into the semicircular canals, to allow for stimulation of the vestibular system via the ampullary nerves.

Participants will be provided with a Sonnet 2 speech processor, used during CI rehabilitation and daily outside the hospital. During VCI experiments, participants will exchange their clinical Sonnet 2 speech processor for a research VCI processor: the audio-motion processor (AMP; Fig 5B). The AMP contains an audio processor unit, and two inertial measurement units, which are housed in the coil and in the audio-processor connector. The AMP includes a base-unit, which has been designed as a research platform to provide a user interface to select various experimental setups, and to connect to the research software (AmpFit). The software packages and all equipment are developed and provided by MED-EL (*Innsbruck*, *Austria*).

## 2.4 Outcome measures

Extensive outcome testing scheduled over the course of the study period, listed in Fig 1, is categorized into main primary outcomes, additional primary outcomes, and secondary outcomes, matching the wide range of objectives of the VertiGO! trial. By measuring outcomes with no vestibular stimulation at baseline (vestibular electrodes turned off, cochlear electrodes turned

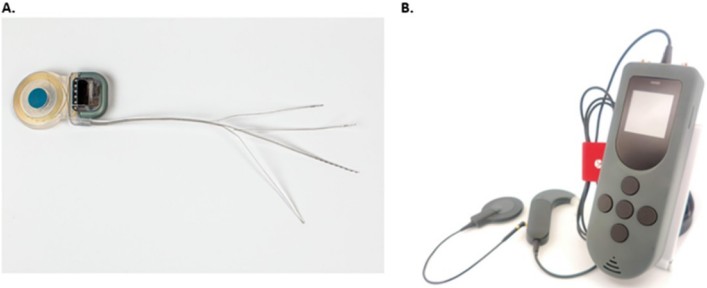

**Fig 5.** (A) The multichannel VCI prototype. (B) The Audio-Motion Processor (AMP) with coil. Reprinted from [43] under a CC BY license, with permission from the SWISS Foundation for Innovation and Training in Surgery and MED-EL, original copyright 2023.

on) and during all three prolonged stimulation periods (vestibular and cochlear electrodes turned on), outcome values will be compared within four program versions. Moreover, by using a test–retest scheme within each prolonged stimulation period, this trial aims to investigate the potential effect of prolonged stimulation within one stimulation paradigm.

**2.4.1 Main primary outcomes: Functional efficacy and safety endpoints.** The main outcome measure is gaze stabilization based on dynamic visual acuity measures (DVA). DVA-testing evaluates reading ability during functional daily life head movements (i.e. while walking on a treadmill), which may be disturbed by oscillopsia in patients with bilateral vestibulopathy [23]. Reduction of oscillopsia was frequently mentioned by BV patients when asked about their expectations of a vestibular implant [44], underlining the clinical importance of this symptom. While standing still and walking on a treadmill, patients will be shown Sloan optotypes (CDHKNORSVZ), which will be projected on a computer screen and presented one by one at different letter sizes (logarithm of the Minimum Angle of Resolution, logMAR). The letter size is defined by using a two-down one-up adaptive staircase procedure (adapted from [45]). The walking speeds on the treadmill include a preferred and maximum walking speed [19]. The outcome value will be defined as the mean logMAR value of the last ten reversal points. This value reflects the functional capability of the vestibular implant to restore visual acuity in a functional setting. The experimental setup is calibrated and validated by comparison with the visual acuity testing at the department of ophthalmology.

As safety outcome measures, all serious adverse events (SAEs), adverse events (AEs) or any other undesired side effects (e.g. psychological burden) in relation to VCI stimulation will be collected and analyzed. Participants will be asked in person about any AE before every study visit. In case an AE/SAE is reported, the flowchart provided by the medical device regulation will be completed: description, severity, relation to investigational procedure, actions taken, treatment and final result.

**2.4.2 Additional primary outcomes.** *2.4.2.1 Vestibular testing.*

<u>Functional head impulse test</u>

Another functional aspect of the high-frequency VOR will also be assessed by the functional head impulse test (fHIT). This test evaluates reading ability during fast head impulses [46]. As the investigator provides a head impulse, an optotype is shortly (80ms) presented on a screen. Participants will be asked to choose the correct optotype out of eight options. The outcome value is the percentage of correct answers for leftwards and rightwards head impulses [47].

<u>Video head impulse test and rotatory chair</u>

In order to quantify the frequency range of the VOR, eye movements will be measured in response to high-frequency and mid-frequency head movements. This will be done by the 3D video head impulse test (vHIT) [48] and the velocity-controlled rotatory chair test. Outcome parameters are VOR gain (video head impulse test and rotatory chair) and saccadic patterns (video head impulse test).

<u>Self-motion perception</u>

Perceptual self-motion thresholds will be measured using a hydraulic perception platform, with 12 degrees of freedom. The procedure will be adapted from procedures previously described [49, 50].

<u>Others</u>

Caloric testing and vestibular evoked myogenic potentials (VEMPs) are two tests that will be used to investigate the residual vestibular function. Caloric testing will be performed during the screening phase according to the procedure, previously discussed [50]. Cervical vestibular evoked myogenic potentials (cVEMPs) and ocular vestibular evoked myogenic potentials (oVEMPs) are measurements of the vestibulo-collic and VOR responses, respectively. Both tests will be performed according to the procedure previously described [50]. The

electromyographic results will be collected before and after surgery to observe any effect of VCI implantation on residual vestibular function.

*2.4.2.2 Gait analysis.* In contrast with the rest of the trial which will follow a single blind procedure, this part of the trial will be conducted as a triple blind procedure, whereby not only are the patients blind to the stimulation mode being applied, but also the gait and balance assessors and the researchers performing the data processing and statistical analysis for the gait and balance parameters will be blind to the stimulation condition. Gait analysis will be conducted on a dual-belt force plate-instrumented treadmill on a motion platform using motion capture and accelerometer data. Participants will walk unperturbed and perturbed (continuous, pseudorandom mediolateral platform sways) at multiple speeds (0.6m/s, 0.8m/s and 1.0m/s) similar to our previous work [51]. The primary parameters of interest are the coefficients of variation of step length, step time, step width and double support time. Additionally, physical therapists will assess patients on the Mini-Balance Evaluation Systems Test (Mini-BESTest) [52] twice per prolonged stimulation period.

**2.4.3 Secondary outcomes.** *2.4.3.1 Telemetry outcomes.* Impedance measurements act as a general safety and functionality check of the device. These measurements may detect problems with the structural integrity of the implant, as well as potential alterations in the tissue surrounding the electrode array, and will therefore be performed multiple times during surgery and before every fitting and test procedure throughout the trial. VI stimulation requires longer phase durations compared to CI stimulation, which might affect impedances and actual compliance limits. Therefore, both standard impedances as well as phase dependent impedance will be measured.

Electrically evoked compound action potentials (ECAPs), using combinations of both cochlear and vestibular electrodes as stimulating and recording probes, will be measured during surgery and at certain follow-up visits. These ECAPs will offer insights into electrical cochlear and vestibular nerve responses as well as the effect of spread of current throughout the inner ear [14]. Secondly, the role of ECAPs in perioperative electrode positioning and postoperative fitting will be explored.

*2.4.3.2 Central vestibular response outcomes.* To assess the response of the central nervous system to VCI stimulation, electrophysiological measurements will be performed, targeting the auditory and vestibular brainstem response. The auditory brainstem response (ABR) is well known and widely used in audiological assessment. The vestibular equivalent, the vestibular brainstem response (VBR), is subject of research but not yet used in clinical practice [53]. The responses can be evoked using the VCI, stimulating locally in the ampulla directly next to the sensory epithelium, giving the opportunity to study the electrically evoked responses (eABR and eVBR). The feasibility of measuring the eVBR will be tested in a setup equivalent to the clinically used eABR evoked with a CI [54]. This might pave the way for further research to characterize the response, and potentially relate it to input parameters and outcome measures such as electrically evoked eye movements. Responses after stimulation will be measured on combinations of all three vestibular electrodes and a selection of the cochlear electrodes. Intersubject variability and the reproducibility of the traces will be used as the main outcome measures. Altogether, this not only gives insights into the processing of vestibular information by the vestibular nerve and related brainstem areas, but this concept might also open doors for applications such as VI fitting and intra-operative electrode positioning.

*2.4.3.3 Auditory outcomes.* Complementary to the outcome measures focusing on the vestibular electrodes, a set of auditory outcome measures will be included to assess functional aspects of combined cochlear and vestibular stimulation. The overall hearing performance of the cochlear electrodes of the VCI will be assessed and followed over time by testing speech recognition using a consonant-nucleus-consonant (CNC) hearing test [55] and a speech-in-

noise test, more specifically the digits-in-noise (DiN) test [56, 57]. By performing tests in situations with VI electrodes on and off and in both static and dynamic settings, potential bidirectional cochlear- and vestibular electrode interactions on functional hearing outcomes will be investigated.

*2.4.3.4 Imaging outcomes.* Prior to VCI insertion, high-resolution temporal bone CT and MRI will be used to determine the patency of the semicircular canals for surgical insertion. Furthermore, the vestibular nerve at the cerebellopontine angle will be evaluated. During surgery, these preoperative imaging samples will be fused with the perioperative imaging (see 2.2.1) to assist with the precise electrode positioning. Post-operative CBCT scans will be performed and analyzed as previously described in [58] using the open source 3D Slicer 4 package [59], to monitor any VCI electrode migration over time.

*2.4.3.5 Patient-reported outcomes.* Both validated questionnaires and semi-structured qualitative interviews will be included to assess a wide range of patient-reported outcome measures (PROM). The set of questionnaires will include the falls efficacy scale international (FES-I) [60], the dizziness handicap inventory (DHI) [61], the oscillopsia severity questionnaire (OSQ) [23, 62], the hospital anxiety and depression scale (HADS) [63], the bilateral vestibulopathy questionnaire (BVQ) [64], and a daily questionnaire with a custom-made set of VAS based questions to evaluate constructs like imbalance, oscillopsia, other physical symptoms, cognitive symptoms, emotional symptoms, limitations and behavioral changes and social life. To allow for calculation of quality-adjusted life years (QALYs) related to receiving a VCI, the euroqol five-dimensional questionnaire (EQ-5D-5L) [65], the ICEpop capability measure for adults questionnaire (ICECAP-A) (validated, Dutch version [66]), and the health utility index mark 3 (HUI-3) [67, 68] will be included to evaluate the construct quality of life. In order to quantify subjective hearing performance, the speech spatial and qualities of hearing scale (SSQ-12) [69, 70] will be included. The potential change in tinnitus burden will be monitored by administering the tinnitus questionnaire (TQ) [71]. The patient-specific functional scale (PSFS) [72] will be used to set and monitor patient-specific training goals during the supervised exercise program. As stated in Fig 1, PROM assessment will be both prior to and after clinical assessments. The order of administration will not be standardized. The mode of administration will be on paper or electronic, both in clinic. No proxy assessment will be used. All questionnaires will be administered in Dutch. The semi-structured interviews aim to include patient-specific experiences related to the VCI, including all biopsychosocial facets. Moreover, VCI-related expectations and the potential influence of the VCI on quality of life will be discussed.

## 2.5 Sample size calculation

Sample size calculation is based on the primary study outcome: DVA score (see 2.4.1). Visual acuity (VA) can be expressed in several different units. However, for data analysis it is important to use the logarithm of the minimum angle of resolution (logMAR), since this unit scales linearly with the geometric progression of the visual acuity chart [73]. Data from previous measurements on the influence of vestibular stimulation on DVA was used for the sample size calculation [19]. Although this data was obtained with a different version of VCI (which did not yet allow simultaneous VI and CI stimulation), it is the closest analog available to approximate the results of this trial. To account for the between-patient difference in VA, the data from the previous experiment, expressed in logMAR, were normalized to the static VA measured for each patient. This yields mean logMAR values of 0.065 (SD 0.061) and 0.205 (SD 0.066) for VI active and VI inactive, respectively. Since the size of the dataset is too small to determine whether or not the data is normally distributed, the sample size was calculated for a

two-sided Wilcoxon signed-rank test. Significance level (alpha) was set at 0.05 and power was set at 0.80 [74, 75]. Both effect size and minimal sample size were calculated with G*Power 3.1.9.7 [76]. Effect size was determined to be 2.20, which resulted in a minimum sample size of five. Given the available resources (e.g. prototype hardware and software, budget limitations due to high (surgical) costs of the procedures) and to provide this statistical background with a safety margin, this trial will aim to include a minimum of eight, with a maximum of 13, patients. Although this is still a relatively small number, both safety and efficacy of the new prototype VCI is believed to be investigated thoroughly with this sample size due to the high frequency of measurements and a total follow-up duration of 5 years.

## 2.6 Statistical methods, data reporting and analysis

The drop-out rate is expected to be very limited in this trial, since patients are extensively counseled before inclusion, closely followed during their standard CI rehabilitation trajectory and contacted before and after each VI related study visit. Apart from the yearly follow-up visits, all visits will be scheduled shortly after each other. Therefore, the drop-out rate is estimated to be zero. However, frequency and reasons for drop-out will be recorded.

All anonymized data will be stored and secured in the digitalized data management software, provided by the local ethical committee (i.e. MACRO). From this software, datasheets will be exported before data cleaning and initial analysis is performed using R [77]. An interim analysis of the data will be performed after a minimum of five VI fitting periods using R. Final statistical analysis will be performed using SPSS 29.0.

Due to the small size, it is expected that the assumptions for parametric tests will not be met for most datasets. To evaluate the effect of VI stimulation in the main and additional primary outcomes (i.e. DVA, fHIT, 3D HIT, Rotatory chair, self-motion perception, Mini-BESTest and CAREN-testing), repeated measures will be collected with multiple within-factors being present. Per subset of data, the median, first and third quartiles will be presented. It is expected that the Friedman test will be the most suitable test to further evaluate statistical significant differences in the datasets. Wilcoxon signed-rank test is expected to be the most suitable test for post hoc analysis. The statistical analysis of the auditory outcomes will be similar. Analysis will be performed in all randomized participants, as-randomized In case of missing data, the reason for missing will be reported for each stimulation mode and compared. No imputation methods will be used. Formal adjustment will be performed in case of multiplicity issues.

Questionnaire outcomes will be analyzed similarly to the primary outcomes. The semi-structured interview will coded and analyzed thematically, following the criteria of Braun & Clarke.

## 2.7 Ethics

Health risks specifically associated with study participation are limited to risks related to the surgery, which are similar to conventional CI surgery, and exposure to ionizing radiation from perioperative fluoroscopy and CT-like imaging and postoperative CBCT. The maximum calculated dose was chosen in consideration of surgical and investigational yield, not exceeding risk category IIa of the International Committee of Radiological Protection (between 0.1 and 1 mSv). Participants will be required to devote time, effort and attention to the study. Participants agree and understand that vestibular stimulation is only applied within the hospital and is not of direct added value in their home situation. Written informed consent was obtained, witnessed by a medical doctor (BLV or BV), using an informed consent form (S1 Appendix, Supporting information, model consent form). Participants will be compensated for their additional traveling expenses and traveling-related burden will be limited by offering a hotel

room close to the hospital, during fitting and testing weeks. Post-trial care will be provided, if applicable, conform local ethical committee standards.

## 3 Discussion

The VertiGO! Trial is the first human trial to assess prolonged efficacy and safety of a multi-channel combined vestibulocochlear implant in a rigorous clinical trial setting. The presented study design enables systematic investigation of a wide range of fundamental-, functional- and patient-reported outcomes. This includes both investigation of the reproducibility of already investigated outcomes (e.g. VOR) and proof of principle of new outcomes (e.g. perceptual self-motion thresholds). The proposed trial aims to structurally evaluate every step of the VCI process, before and during stimulation. First, by improving electrode positioning by using both perioperative fluoroscopy and 3D images fused with pre-operative imaging. Secondly, by evaluating different fitting strategies. This trial allows for the evaluation of prolonged processes, by measuring the induced responses at multiple time points before, during and after prolonged VCI stimulation with different types of modulation. DVA will serve as the main primary outcome for VI efficacy, because it provides objective measurements of the degree of functional restoration of the VOR, directly related to patient complaints and symptoms. Besides DVA, the extensive test battery in this trial will help in choosing the most applicable outcome parameters for future confirmatory trials with larger populations. Moreover, to determine the potential value of the VCI, an early health technology assessment will depend on these outcome parameters as well.

### 3.1 Development of a clinically usable VI fitting strategy

The approach proposed in this trial is expected to obtain essential knowledge about VI fitting. This approach aims to develop a fitting method, which can be translated into clinical practice (e.g. timing of sessions, setups). At this point, no data is present to build a clinical fitting scheme. By investigating the influence of different variables on DR, such as (level of) baseline stimulation, signal waveform (i.e. square or sine), pulse characteristics (i.e. phase duration, rate) and time (i.e. development over time after surgery), fitting will be further clarified. After initial determination of the DR, different modulation settings can be used. By varying the encoding of head movements (input) in modulations of the baseline signal (output), the effect of modulation settings can be investigated throughout the DR. Although this trial aims for a standardized fitting protocol draft, it might be possible or even essential to develop personalized fitting strategies. By comparing individual DR values structurally in these different paradigms, interindividual differences will be investigated further.

### 3.2 Potential CI—VI interactions and the effects of interleaved stimulation

Since vestibular structures are position close to the cochlear structures [78], it is possible that the currents delivered by the CI electrodes spread to the surrounding vestibular structures and that the same applies for currents delivered by the VI electrodes to the auditory structures. Secondly, in order to enable both cochlear and vestibular stimulation, specific auditory gaps are introduced in the classical auditory stream [79]. These auditory gaps, which are relatively large compared to the pulse duration of regular CI stimulation, are inevitable when sequential stimulation of both vestibular and cochlear electrodes is addressed within one stimulation frame. Without these gaps, it would not be possible to stimulate both vestibular and cochlear electrodes concurrently [79]. The spread of excitation and the gaps in the auditory stream might influence CI and VI performance bidirectionally. The interaction effect of brief trains of combined stimulation of the cochlear and vestibular arrays of the VCI was investigated previously

[80]. Although this study only showed fundamental results from a small sample size, it suggests that interleaved stimulation might elicit potentially clinically relevant changes in electrically evoked eye velocity, loudness percept and pitch percept [80]. However, introducing specific gaps and distributions in the auditory stream for a cochlear implant did not show deterioration of speech perception [79].

It is known that CI stimulation possibly impacts vestibular test results [81]. Recently, this phenomenon was used to investigate possible vestibular improvement in children. A significant reduction of the number of falls and a better postural control were measured in conditions where bilateral head-referenced CI feedback was provided [82, 83]. However, at this point it is unclear whether co-stimulation and/or restored auditory cues contributed most to the improved balance function.

These results underline the necessity to evaluate the potential interactions between VI and CI more extensively on a clinically relevant level. By including an auditory test set during the prolonged stimulation periods and during follow-up visits, the CI performance can be compared between the regular CI processor versus the VCI research processor with and without VI stimulation turned on.

## 3.3 Limitations

An eminent problem when implementing a crossover design in order to evaluate VCI performance is the occurrence of a first-order carry-over effect [84]. However, previous studies with repeated acute stimulations showed stable results over time with as little as one day of wash-out in between [16–18, 39, 85]. Moreover, the wash-out between each stimulation period is chosen to be at least three days to further decrease the change of carry-over effects.

Previously, a decrease in electrically elicited eye movement responses was measured after prolonged stimulation in both animal and human studies [86–88]. This might be due to a suppression at vestibular neuron synapses, resulting in a decrease in neuronal sensitivity [86]. At this point, the timing and degree of this suppression are not clear. Therefore, it is possible that changing degrees of suppression of eye movements are present during different stages within this trial. Due to the limited duration of 4 days of prolonged stimulation per stimulation mode, it is possible that a steady state phase is not yet reached within one period of prolonged stimulation.

Due to the small sample size, coupled with the range of etiologies and degrees of residual function seen in patients, the outcomes of this study may not serve as proof for population-wide claims. This trial focuses on proof related to individual efficacy. In this way, a variety of patient characteristics (e.g. etiology, duration of vestibular function) will be investigated. However, due to the use of extensive and strict inclusion criteria, it is challenging to enroll participants out of the full spectrum of BV etiologies. Although the sample size is small, the crossover design and multiple follow-up visits facilitate extensive data collection from the limited number of participants. Also, this trial aims to apply insights at an individual level to a stimulation strategy on group level. More specifically: individual fitting data will be analyzed, fittings will be optimized, and implemented before prolonged stimulation of the whole sample is scheduled.

Another limitation in external validation of the results is the hospital setting of our experiments. Although the possibilities within the hospital are limited, an out-of-hospital activation pattern will be mimicked as much as possible. This will be realized by including exercise programs and by encouraging participants to move and behave 'as normally as possible'. The exercise program included in every prolonged stimulation period aims to activate participants and prevent any movement avoidance during stimulation. By using sets of progressive exercises,

based on the updated clinical practice guideline [89], a first experience in the possibilities of vestibular rehabilitation in a VCI-patient population will be gained. Multicenter trials with increased sample sizes and stimulation in an out-of-hospital context will be interesting follow-up projects, in order to gain more data about the functional impact on daily life.

### 3.4 Early health technology assessment and future product development

To inform all stakeholders about the potential value of the implant and steering the innovation of cochleovestibular stimulation in an early phase, an early health technology assessment is required [90]. The most relevant outcome measures to evaluate the spectrum of expected performance of the VCI, will be identified in this trial. Changes in participant-reported disability and quality of life have already been reported after prolonged stimulation with a VI-only device [28]. Based on these findings and other previously reported perspectives in a BV patient population [44], it is expected that anticipated value by patients plays a crucial role in the early assessment of the usability of the device. Further assessment of the impact of the new VCI technology on the individual patient in their home-situation, their environment and even society as a whole will be planned during follow-up studies. These studies might help to define candidacy criteria in terms of etiologies causing vestibulopathy and/ or hearing loss.

## 4 Conclusion

The VertiGO! Trial is the first human trial to assess prolonged efficacy and safety of a multi-channel combined vestibulocochlear implant in a rigorous trial setting. The presented study design enables systematic investigation of VI-fitting strategies and a wide range of fundamental-, functional- and patient-reported outcomes.

## Supporting information

**S1 Appendix. Model consent form.**
(DOCX)

**S2 Appendix. SPIRIT checklist.**
(DOCX)

**S3 Appendix. World Health Organization trial registration data set.**
(DOCX)

**S1 Protocol.**
(PDF)

## Author Contributions

**Conceptualization:** Bernd L. Vermorken, Benjamin Volpe, Stan C. J. van Boxel, Joost J. A. Stultiens, Marc van Hoof, Rik Marcellis, Alexander van Soest, Nils Guinand, Angélica Pérez Fornos, Elke Devocht, Raymond van de Berg.

**Methodology:** Bernd L. Vermorken, Benjamin Volpe, Stan C. J. van Boxel, Joost J. A. Stultiens, Marc van Hoof, Rik Marcellis, Alexander van Soest, Chris McCrum, Kenneth Meijer, Nils Guinand, Angélica Pérez Fornos, Elke Devocht, Raymond van de Berg.

**Project administration:** Bernd L. Vermorken.

**Supervision:** Kenneth Meijer, Angélica Pérez Fornos, Vincent van Rompaey, Elke Devocht, Raymond van de Berg.

**Visualization:** Bernd L. Vermorken.

**Writing – original draft:** Bernd L. Vermorken.

**Writing – review & editing:** Bernd L. Vermorken, Benjamin Volpe, Stan C. J. van Boxel, Joost J. A. Stultiens, Marc van Hoof, Elke Loos, Chris McCrum, Nils Guinand, Angélica Pérez Fornos, Vincent van Rompaey, Elke Devocht, Raymond van de Berg.

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
