## [Decision Letter · Decision Letter 0]

7 Jan 2024

PONE-D-23-33489The VertiGO! Trial protocol: a prospective, single-center, patient-blinded study to evaluate efficacy and safety of prolonged daily stimulation with a multichannel vestibulocochlear implant prototype in bilateral vestibulopathy patientsPLOS ONE

Dear Dr. Vermorken,

Thank you for submitting your manuscript to PLOS ONE. After careful consideration, we feel that it has merit but does not fully meet PLOS ONE’s publication criteria as it currently stands. Therefore, we invite you to submit a revised version of the manuscript that addresses the points raised during the review process.

We look forward to receiving your revised manuscript.

Kind regards,

Renato S. Melo, PhD

Academic Editor

PLOS ONE

4. We note that Figure 5 in your submission contain copyrighted images. All PLOS content is published under the Creative Commons Attribution License (CC BY 4.0), which means that the manuscript, images, and Supporting Information files will be freely available online, and any third party is permitted to access, download, copy, distribute, and use these materials in any way, even commercially, with proper attribution. For more information, see our copyright guidelines: http://journals.plos.org/plosone/s/licenses-and-copyright.

1. You may seek permission from the original copyright holder of Figure 5 to publish the content specifically under the CC BY 4.0 license.

5. We note that the original protocol that you have uploaded as a Supporting Information file contains an institutional logo. As this logo is likely copyrighted, we ask that you please remove it from this file and upload an updated version upon resubmission.

Reviewers' comments:

Reviewer's Responses to Questions

**Comments to the Author**

1. Does the manuscript provide a valid rationale for the proposed study, with clearly identified and justified research questions?

Reviewer #1: Partly

Reviewer #2: Partly

2. Is the protocol technically sound and planned in a manner that will lead to a meaningful outcome and allow testing the stated hypotheses?

Reviewer #1: No

Reviewer #2: Yes

3. Is the methodology feasible and described in sufficient detail to allow the work to be replicable?

Reviewer #1: Yes

Reviewer #2: No

4. Have the authors described where all data underlying the findings will be made available when the study is complete?

Reviewer #1: Yes

Reviewer #2: Yes

5. Is the manuscript presented in an intelligible fashion and written in standard English?

Reviewer #1: Yes

Reviewer #2: Yes

6. Review Comments to the Author

You may also provide optional suggestions and comments to authors that they might find helpful in planning their study.

Reviewer #1: In this study protocol, a single-blind randomized crossover controlled design with three time periods is being proposed to investigate the safety and efficacy of prolonged daily motion modulated stimulation with a multichannel VCI prototype. The expected target sample size is 8 to 13.

Major revision:

The statistical analysis plan needs to be fully fleshed out. The sample size is too small to test for normality of data; therefore, nonparametric statistical methods are recommended (Line 501). For each objective, identify the describe the nonparametric summary results (medians, first and third quartiles, ranges, etc.) and tests that will be applied.

Minor revisions:

1- Line 373: Indicate if adverse events will be collected according to a standardized method.

2- Identify the software that will be used to capture the data as well as the software that will be used for the statistical analysis.

Reviewer #2: From the paragrah 58- "Bilateral vestibulopathy (BV) is defined as a severe loss of function of both balance organs, which 59 represents a major handicap involving strong balance disturbances, higher risk of falling, oscillopsia (i.e. a 60 symptom of blurred vision during head movements), and associated loss of autonomy and quality of life" this definition of BV it is not enough, even when in Methods, the authors provide quantitative measurements, we suggest an accepted international definition, Bárány´s Society, for example.

Regarding the number of patients even when with 5 the protocol can get an Statitical Power of 80, the authors should stimate the drop-out rate.

7. PLOS authors have the option to publish the peer review history of their article (what does this mean?). If published, this will include your full peer review and any attached files.

Reviewer #1: No

Reviewer #2: **Yes: **Sergio Carmona

---

## [Author Response · Author response to Decision Letter 0]

11 Jan 2024

We thank the reviewers for their comments on the manuscript and have edited the manuscript to address their concerns. Please find below an overview of the comments and our replies (italics). 

Reviewer 1 

Major revision: 

The statistical analysis plan needs to be fully fleshed out. The sample size is too small to test for normality of data; therefore, nonparametric statistical methods are recommended (Line 501). For each objective, identify the describe the nonparametric summary results (medians, first and third quartiles, ranges, etc.) and tests that will be applied. 

We agree upon the limited outline of the statistical plan. Therefore, more information was added in paragraph 2.6. 

Minor revisions: 

1- Line 373: Indicate if adverse events will be collected according to a standardized method.

2- Identify the software that will be used to capture the data as well as the software that will be used for the statistical analysis. 

1 – A description of the standardized method was added at the end of paragraph 2.4.1. 

2 – Software details were added in paragraph 2.6 

Reviewer 2 

From the paragrah 58- "Bilateral vestibulopathy (BV) is defined as a severe loss of function of both balance organs, which 59 represents a major handicap involving strong balance disturbances, higher risk of falling, oscillopsia (i.e. a 60 symptom of blurred vision during head movements), and associated loss of autonomy and quality of life" this definition of BV it is not enough, even when in Methods, the authors provide quantitative measurements, we suggest an accepted international definition, Bárány´s Society, for example. 

A reference to the Barany’s Society definition was added, together with clarifying the statement in the Introduction. 

Regarding the number of patients even when with 5 the protocol can get an Statitical Power of 80, the authors should stimate the drop-out rate. 

The drop-out rate is estimated to be zero, due to the frequent contact moments between participants and researchers. Participant retention will be promoted. Some background information about this estimated drop-out rate was added in paragraph 2.6

Additional requirements 

1. Style requirements 

Style was adapted, conform requirements. 

2. Additional details regarding participant consent were added in the Methods – Ethics section (paragraph 2.7) and in the online submission information. 

3. Grant information in ‘Funding Information’ and ‘Financial Disclosure’ were checked and matched.

4. Consent to the CC BY 4.0 License was provided for Fig 5. The documents were attached labeled as ‘Content Permission Form’ as ‘Other’. 

Figure caption of Fig 5 was edited. 

5. An updated version of the original protocol was uploaded, without copyrighted images/logo. 

6. The caption S1 Appendix was added to the Supporting Information at the end of the manuscript, one in-text citation was updated (paragraph 2.1). 

7. Figure files were uploaded to PACE.

---

## [Decision Letter · Decision Letter 1]

11 Mar 2024

The VertiGO! Trial protocol: a prospective, single-center, patient-blinded study to evaluate efficacy and safety of prolonged daily stimulation with a multichannel vestibulocochlear implant prototype in bilateral vestibulopathy patients

PONE-D-23-33489R1

Dear Dr. Vermorken,

We’re pleased to inform you that your manuscript has been judged scientifically suitable for publication and will be formally accepted for publication once it meets all outstanding technical requirements.

Kind regards,

Renato S. Melo, PhD

Academic Editor

PLOS ONE

Additional Editor Comments (optional):

Reviewers' comments:

Reviewer's Responses to Questions

**Comments to the Author**

1. Does the manuscript provide a valid rationale for the proposed study, with clearly identified and justified research questions?

Reviewer #1: Yes

Reviewer #2: Partly

2. Is the protocol technically sound and planned in a manner that will lead to a meaningful outcome and allow testing the stated hypotheses?

Reviewer #1: Yes

Reviewer #2: Yes

3. Is the methodology feasible and described in sufficient detail to allow the work to be replicable?

Reviewer #1: Yes

Reviewer #2: Yes

4. Have the authors described where all data underlying the findings will be made available when the study is complete?

Reviewer #1: No

Reviewer #2: Yes

5. Is the manuscript presented in an intelligible fashion and written in standard English?

Reviewer #1: Yes

Reviewer #2: Yes

6. Review Comments to the Author

You may also provide optional suggestions and comments to authors that they might find helpful in planning their study.

Reviewer #1: All comments have been adequately addressed.

Reviewer #2: Even with an small sample the VertiGo protocol is a promising investigation about the utility of vestibular implants in the real world

7. PLOS authors have the option to publish the peer review history of their article (what does this mean?). If published, this will include your full peer review and any attached files.

Reviewer #1: No

Reviewer #2: No

---

## [Editor Report · Acceptance letter]

18 Mar 2024

PONE-D-23-33489R1 

PLOS ONE

Dear Dr. Vermorken, 

I'm pleased to inform you that your manuscript has been deemed suitable for publication in PLOS ONE. Congratulations! Your manuscript is now being handed over to our production team.

Kind regards, 

on behalf of

Dr. Renato S. Melo 

Academic Editor

PLOS ONE